# Harnessing the LdCsm RNA Detection Platform for Efficient microRNA Detection

**DOI:** 10.3390/ijms24032857

**Published:** 2023-02-02

**Authors:** Zhenxiao Yu, Jianan Xu, Qunxin She

**Affiliations:** CRISPR and Archaea Biology Research Center, State Key Laboratory of Microbial Technology and Microbial Technology Institute, Shandong University, Qingdao 266237, China

**Keywords:** CRISPR-Cas10, type III-A, HD domain, DNase, RNA detection, miRNA, modification

## Abstract

In cancer diagnosis, diverse microRNAs (miRNAs) are used as biomarkers for carcinogenesis of distinctive human cancers. Thus, the detection of these miRNAs and their quantification are very important in prevention of cancer diseases in human beings. However, efficient RNA detection often requires RT-PCR, which is very complex for miRNAs. Recently, the development of CRISPR-based nucleic acid detection tools has brought new promises to efficient miRNA detection. Three CRISPR systems can be explored for miRNA detection, including type III, V, and VI, among which type III (CRISPR-Cas10) systems have a unique property as they recognize RNA directly and cleave DNA collaterally. In particular, a unique type III-A Csm system encoded by *Lactobacillus delbrueckii* subsp. *bulgaricus* (LdCsm) exhibits robust target RNA-activated DNase activity, which makes it a promising candidate for developing efficient miRNA diagnostic tools. Herein, LdCsm was tested for RNA detection using fluorescence-quenched DNA reporters. We found that the system is capable of specific detection of miR-155, a microRNA implicated in the carcinogenesis of human breast cancer. The RNA detection system was then improved by various approaches including assay conditions and modification of the 5′-repeat tag of LdCsm crRNAs. Due to its robustness, the resulting LdCsm detection platform has the potential to be further developed as a better point-of-care miRNA diagnostics relative to other CRISPR-based RNA detection tools.

## 1. Introduction

Developing efficient RNA detection platforms is crucially important for the diagnosis of human diseases such as cancers and infectious diseases caused by RNA viruses. For the latter, the eruption and spreading of human immunodeficiency virus (HIV) [1], SARS coronavirus, MERS coronavirus [2], Zika virus [3], Ebola virus [4], and the recent epidemic of SARS-CoV-2 [5] have been catastrophes to human beings in the whole world; for the former, human cancers claim millions of death and account for a large amount of the human healthcare spending every year [6,7]. Since small RNAs, such as circular RNAs (circRNAs) and microRNAs (miRNAs), often serve as biomarkers for various cancer diseases [8,9,10], developing accurate diagnostics of these human disease-related RNAs as well as human RNA viruses is crucially important to the prevention of these human diseases.

Currently, quantitative reverse transcriptase-polymerase chain reaction (qRT-PCR) still serves as the traditional gold standard for RNA detection [11,12,13], but the method is relatively time-consuming and requires thermocyclers, which prevents its application in the point-of-care (POC) diagnosis. RNA detection platforms based on various isothermal amplification methods provide efficient alternatives to qRT-PCR. The means of target amplification include reverse transcription-recombinase polymerase amplification [14,15], reverse transcription-loop-mediated isothermal amplification [16,17], rolling circle amplification (RCA) [18], catalytic hairpin assembly (CHA) [19,20,21], and NEase-based amplification [22,23]. However, since these amplification methods are untargeted, small RNA species are often underrepresented in the amplified samples; in addition, as the amplification processes are error-prone, it is currently unknown how the errors generated during amplification could influence the detection results [24,25,26].

Clustered Regularly Interspaced Short Palindromic Repeat (CRISPR)-associated systems (CRISPR-Cas) provide the prokaryotic adaptive immunity [27]. This immune system encodes ribonucleoprotein complexes consisting of mature CRISPR (cr)RNAs and Cas proteins, in which the crRNA specifically recognizes the invading nucleic acid by sequence complementarity, whereas the Cas proteins silence the invaders by nucleic acids cleavage [28,29,30] and/or induction of abortive infection [31]. Due to their programmability, CRISPR-Cas systems have been exploited for gene editing in all three domains of life [32]. In addition, ribonucleoprotein complexes of some CRISPR systems exhibit trans nucleic acid (NA) cleavage, an activity that is also termed as the collateral RNA/DNA cleavage [33,34,35]. Since the trans NA cleavage is to be activated either by an RNA or DNA showing sequence complementarity to their crRNA, which is also called the protospacer [36], this unique feature has rendered it possible to develop nucleic acid detection platforms based on these CRISPR-Cas systems. To date, nucleic acid detection tools have been developed with the DNA-activated DNase of the CRISPR-Cas12 systems [34,37], the RNA activated RNase of the CRISPR-Cas13 systems [33], and RNA-activated cOA (cyclic oligoadenylate) signal transduction pathway as well as the RNA-activated DNase of several CRISPR-Cas10 systems [38,39,40,41]. 

In addition, Cas12 and Cas13 effectors have been explored for miRNA detection, but each type of detection tool exhibits some drawbacks. For Cas12-based miRNA detection, miRNA targets have to be converted to corresponding DNAs, which activate Cas12 for cleavage of a DNA reporter. The employed methods of conversion include the rolling circle amplification [42], the catalytic hairpin assembly [43], the cascade strand displacement reaction [44] or the hybridization chain reaction [45]. This is different from the Cas13 miRNA detection platform, in which the RNA-activated RNase recognizes miRNA directly [46,47]. However, the collateral RNase of Cas13 also destructs miRNAs to be detected, which could have a profound influence on miRNA quantification. As for Cas13, CRISPR-Cas10 effectors also recognize RNA directly, but none of type III CRISPR-based RNA detection tools have been reported for miRNA diagnosis thus far.

Recently, we characterized a type III-A Csm system in *Lactobacillus delbrueckii* subsp. *bulgaricus* (LdCsm) that exhibits very strong RNA-activated DNase activity [48,49]. In this article, we constructed a LdCsm-based RNA detection tool based on the collateral DNase activity of the antiviral system and optimized its detection efficiency. Compared with the Cas13a-based detection system, the LdCsm-based RNA detection tool performs better in the presence of unspecific RNAs, the reagents are more robust during storage at room temperature (RT), and the detection on miR-155a, a breast cancer-related miRNA, reached 500 pM-2 nM without any form of signal amplification.

## 2. Results

### 2.1. Developing the LdCsm RNA Detection Platform

Previously, we showed that the LdCsm effector is capable of distinguishing a single nucleotide difference between the target RNAs derived from 23S rRNAs of two lactobacilli species [48]. The rationale of the assay is as follows. Firstly, the LdCsm ribonucleoprotein complexes are prepared by expression in, and purification from, *E. coli*. To do that, the type III-A *cas* genes of the *L. delbrueckii* (Ld Cas module) were cloned into a plasmid expression vector, whereas the RNAs to be detected were employed to design spacers for a mini-CRISPR array, which was then cloned to another plasmid vector (Figure 1a); upon transformation into *E. coli* cells, plasmid-borne expression yielded Cas proteins and pre-crRNA transcripts, which eventually formed ribonucleoprotein complexes (Figure 1a). Then, the LdCsm effector was purified and employed in the RNA detection assay in which the crRNA in the III-A effector complex recognizes the specific protospacer RNA by sequence complementarity to form a target RNA-LdCsm ternary complex. During the process, there occurs a conformational change in which the Cas10 HD domain is activated for the collateral DNA cleavage, and the trans-cleavage activity destructs fluorescence-quenched ssDNA reporter molecules, releasing fluorescent products to be detected and quantified by fluorometry [48]. Therefore, if spacers are designed based on RNAs to be detected, such as RNA viruses and microRNAs, the resulting LdCsm effector purified from *E. coli* can be employed to detect their corresponding RNA protospacers (also called target RNAs) since the latter function as RNA activators to the designed LdCsm effectors, whereas their non-homologous RNAs (unspecific RNAs) cannot (Figure 1b).

The trans-cleavage of ssDNA by type III CRISPR effectors basically works on ssDNAs of any sources. Consequently, this immune response has to be carefully controlled, and indeed, type III CRISPR systems have evolved the spatiotemporal regulation to limit its occurrence at the right place and in the right time. Namely, the immunity is triggered by transcription of the target site, whereas it is switched off by the enzymatic destruction of target RNAs by the nuclease hosted in the large backbone subunits, which is Csm3 for III-A systems. 

Previously we showed that LdCsm-dCsm3, a mutated effector that is inactivated for backbone cleavage, is also capable of distinguishing a specific target RNA versus an unspecific one. Herein, the same mutated effector was employed for establishing the LdCsm-based RNA detection platform. Reactions were set up with an equal concentration of target RNA S1 (CTR-full) or S10, a non-homologous RNA, in the presence of a fluorescence-quenched polythymidine (Poly-T) reporter and LdCsm-dCsm3. S1 is the activator RNA that facilitates the cleavage of the Poly-T reporter, yielding fluorescent molecules that can be detected and quantified with a fluorometer. As shown in Figure 2a,b, relative fluorescence units (RFU) increase along with the incubation time in presence of S1, whereas no fluorescent signal is produced in the presence of S10, and these results indicated that the LdCsm-dCsm3 system is capable of specific RNA detection. 

The metal ion dependence of the LdCsm DNase was tested in our previous work where we found that the LdCsm DNase is activated by 10 mM MgCl2, and the addition of 50 mM KCl further increases the DNA cleavage activity [48]. Here, the LdCsm-dCsm3 detection system was optimized for the reaction pH and temperature. The ranges of pH and temperature values tested were 6.4–8.0 and 25–60 °C, respectively. We found that the optimal pH is 7.0 for the LdCsm-dCsm3 enzyme (Figure 2c). In the determination of the optimal temperature, we found it changed along with the change of assay time periods: the DNase activity is the highest at 50–60 °C for the first 10–15 min, but the LdCsm DNase is also inactivated during incubation, whereas incubation at 30–37 °C yields the linear increase of fluorescent products for 30 min, indicative of the occurrence of the optimal DNase reaction during the entire assay period (Figure 2d). These reaction conditions were then adopted for the LdCsm-dCsm3 DNase-based RNA detection. 

The Poly-T reporter was designed based on the preferred sites of cleavage by the *Sulfolobus islandicus* III-B Cmr system [50]. While this DNA reporter functions in the RNA detection with the LdCsm system [48], the nucleotide motif preference of the III-A system remained to be revealed. To do that, seven different 5′ FAM-labeled ssDNAs (sequences listed in Appendix A) were determined for main cleavage sites of LdCsm-dCsm3 by polyacrylamide gel electrophoresis (PAGE) analyses of trans-cleavage products (Figure 3a and Appendix A). The main cleavage sites were then deduced for each ssDNA substrate based on the sizes of cleaved products of a high abundance (Figure 3b). This revealed a total of 39 main cleaved sites of 10 different dinucleotides including 11 CT, 9 CC, and 7 CA dinucleotides, whose sum accounts for 69% of the total number (Appendix A). Moreover, CTs are among the main cleavage sites in all seven ssDNA substrates, while CC and CA occurred in six and four of them, respectively (Appendix A). Therefore, we concluded that CT, CC, and CA dinucleotides are preferred ssDNA sites of the LdCsm-dCsm3 DNase. 

We then designed a CT/CC-rich reporter (CT reporter) and tested it for reporting efficiency, along with the Poly-T reporter, in the fluorescence assay (Figure 3c, sequences also listed in Appendix A). We found that in presence of the equal concentration of target RNA, LdCsm-dCsm3 detection with the CT reporter produced 2–3 fold higher RFU signals than with the Poly-T reporter (Figure 3d,e). Thus, compared to the Poly-T reporter, the CT reporter is more efficient in RNA detection as it elevates the detection efficiency of LdCsm-dCsm3 detection system by 2–3 folds.

### 2.2. Comparison of LdCsm and Cas13a Detection Systems

Cas13 effectors exhibit the target RNA-activated collateral nuclease activity [33], which is in contrast to the trans-cleavage of ssDNA by CRISPR-Cas10 RNPs as described above. Several nucleic acid detection tools have been developed based on Cas13 antiviral systems [33,51,52]. Herein, the efficiency of the LdCsm-dCsm3 detection system was compared with a Cas13a detection system to explore their potential in diagnostic applications. The Cas13a detection platform was based the *Leptotrichia wadei* (Lwa) Cas13 protein (Bio-lifesci, Guangzhou, China), a synthetic S1 crRNA, and a fluorescence-quenched RNA reporter, as listed in Appendix A. 

In clinical diagnosis, samples for nucleic acid detection are usually complex, and the target RNA is often comprised of a very small proportion of total RNAs in the samples. Therefore, it is very important for an RNA detection system to specifically detect the target RNA in the presence of large amounts of unrelated RNAs in the samples, a property that is referred to as “the tolerance to unspecific RNAs”. Herein, the LdCsm-dCsm3 and the Cas13a detection systems were evaluated for this feature. Target RNA of a series of concentrations (0.2–2 nM) was mixed with an unspecific RNA sample, which is the total RNA prepared from *E. coli* cells, to a final content of 2 or 10 ng/µL. The resulting mass ratios of the target RNA vs. unspecific RNAs were 1:670–67 and 1:3330–333 for the two series of reaction, respectively. As shown in Figure 4a, when target RNA was comprised of 1.5% of the total RNA in the sample (1:67), Cas13a retained ca. 26% of the detection activity, but LdCsm retained ca. 80%; when the target RNA composition dropped to 0.3% (1:333), their residual detection efficiency was ca. 5% and 45% for the Cas13a and LdCsm systems, respectively (Figure 4a). Thus, the LdCsm detection system exhibited a much higher level of tolerance to unspecific RNAs than the Cas13a system.

The robustness of miRNA detection platforms is another essential feature, which is especially important for developing the POC diagnostics. We assessed the stability of LdCsm-dCsm3 and Cas13a detection systems after storing these Cas enzymes at different temperatures for up to one month including −80 °C, −20 °C, 4 °C, and RT. Aliquots of Cas13a and LdCsm enzymes were prepared and stored at the above specified conditions, and during the storage, aliquots were taken out and tested regularly for the activity of miRNA detection. We found that both LdCsm-dCsm3 and Cas13a retained the full activity during the entire period of storage at −20 °C or −80 °C. However, Cas13a completely lost the detection ability after storage at 4 °C after the same period of storage, and it rapidly lost the activity at RT (after 7 days); in contrast, after storage at 4 °C or RT for 28 days, LdCsm-dCsm3 still retained 83% and 62% of the detection efficiency, respectively (Figure 4b). Thus, the LdCsm-dCsm3 detection system serves as a more promising candidate for developing POC detection tools relative to the Cas13 detection platform.

### 2.3. Determination of the Length Requirement of Activator RNA in the LdCsm-dCsm3 Detection Platform

Target RNA activation of the DNase activity of LdCsm-dCsm3 has to satisfy two requirements: (1) the target RNA should carry the sequence that is complementary to spacer region of the LdCsm crRNA (this region in target RNA is termed protospacer); (2) the 3′-protospacer-flanking sequence (3′-antitag) of target RNA should show mismatches with the 5′-repeat tag (or 5′-repeat handle) of the LdCsm crRNAs [48]. This kind of target RNA is termed cognate target RNA (CTR). To date, the length of protospacer or mismatching 3′-antitag of target RNA required to fully activate LdCsm-dCsm3 remains unclear, which could affect the detection efficiency relying on its DNase activity. 

To explore the effect of length of target RNA on LdCsm-dCsm3 detection system, activation of the LdCsm-dCsm3 system was tested with RNAs with different lengths of protospacer and/or 3′-antitags (Figure 5a, sequences also listed in Appendix A), with the activity measured by fluorescence DNA reporter cleavage assay. Detection activity (represented by the increasing rate of relative fluorescence units, RFU) of LdCsm-dCsm3. This revealed that CTR with a 22–23 nt protospacer fully activated the LdCsm-dCsm3 DNase, and further reducing the protospacer by a few more nucleotides (17 nt or 19–20 nt) strongly impeded the activation, as the residual activity was 11–25% (Figure 5b). On the other hand, a minimum of 4 nt of mismatched 3′-antitag is required for a CTR to activate LdCsm-dCsm3 (Figure 5c). Taken together, microRNAs of ca. 24 nt should be able to trigger 11–25% DNase activity of LdCsm-dCsm3, and this provided the basis for harnessing LdCsm-dCsm3 system for microRNA detection.

### 2.4. Developing the LdCsm-dCsm3-Based microRNA Detection System

As one of the most common cancer in the world, breast cancer (BC), accounts for over 25% of female cancer cases and causes one of the biggest cancer-related death rolls in women [7]. miR-155 is a 24 nt miRNA found to be deregulated in BC tissues, which could function as a potential biomarker for early diagnosis of BC [53,54,55]. To investigate whether LdCsm-dCsm3 detection system can be applied to sense miRNA, we tested its ability in detecting miR-155. Two spacers including S-155-1 and S-155-2 were designed for construction of expression systems of LdCsm-dCsm3 targeting miR-155. S-155-1 is complementary to 5′ 1–20 nt of miR-155, and S-155-2 is complementary to 5′ 1–19 nt of it (Figure 6a). Then, pUCE-S-155-1 and pUCE-S-155-2 plasmids were constructed and used to express and purify LdCsm-dCsm3 complexes targeting miR-155. 

miR-155 was synthesized by the Azenta Biotechnology Company (Qingdao, China) and used as the target RNA to test detection capability of the LdCsm-dCsm3-155-1 and LdCsm-dCsm3-155-2 effectors. As shown in Figure 6b, both LdCsm effectors cleaved the fluorescence-quenched ssDNA reporter and produced a RFU signal in the presence of miR-155, while no production of any RFU signal was detected in the presence of unspecific RNA S10, indicating that both LdCsm-dCsm3-155-1 and LdCsm-dCsm3-155-2 are capable of specific detection of miR-155. In addition, in the presence of the same amount of miR-155, LdCsm-dCsm3-155-2 generated 3–4-fold higher detection signals than LdCsm-dCsm3-155-1.

As mentioned above, activation of LdCsm-dCsm3 detection system requires a target RNA carrying a 4 nt or longer 3′-anititag showing mismatches with the corresponding 5′-repeat tag of LdCsm crRNAs (Figure 5c). However, for miRNAs of 24 nt, their 3′-end sequences have to be directly employed as the 3′-antitags, as entire miRNAs are too short to provide any alternative 3′-antitags exhibiting full mismatches to the 5′-repeat tag of the corresponding crRNAs. For example, each 5′-repeat tag of crRNA in LdCsm-dCsm3-155-1 or LdCsm-dCsm3-155-2 has a one-base pairing between 3′-antitag of miR-155 (Figure 6a). 

To optimize the detection efficiency of the LdCsm-dCsm3 detection system on miR-155, we chose to introduce single-base substitutions into the repeat sequence of the Ld CRISPR array to generate full mismatches between the 5′-repeat tag of crRNA and the 3′-antitag of miR-155. Examination of the 3′-end of miR-155 revealed that A at position −3 of the 5′-repeat tag is complementary to it in one scenario while the A at position −5 of the 5′-repeat tag is complementary to it in another (Figure 6a). We then introduced point mutation (A→T) to the two positions individually, producing pUCE-S-155-1-A−3U and pUCE-S-155-2-A−5U, respectively. pUCE-S-155-1-A−3U and pUCE-S-155-2-A−5U plasmids were then used to produce LdCsm-dCsm3-155-1-A−3U and LdCsm-dCsm3-155-2-A−5U complexes. The resulting crRNAs then carried crRNAs with the 5′-repeat tags showing full mismatches to the 3′-end of miR-155 (Figure 6a). 

LdCsm-dCsm3-155-1-A−3U and LdCsm-dCsm3-155-2-A−5U were successfully purified, and the detection efficiency of them on miR-155 was compared with LdCsm-dCsm3-155-1 and LdCsm-dCsm3-155-2. Compared to LdCsm-dCsm3-155-1, the detection activity of LdCsm-dCsm3-155-1-A−3U on miR-155 was not improved (Figure 6c). In contrast, the detection activity of LdCsm-dCsm3-155-2-A−5U on miR-155 was 3.9-fold higher than that of LdCsm-dCsm3-155-2, which was evidently enhanced (Figure 6c). As a result, the detection efficiency of LdCsm-dCsm3-155-2 was successfully enhanced by introducing base mutation into its 5′-repeat tag. 

The LoD of the LdCsm-dCsm3-155-2-A−5U effector was determined with miR-155. The RNA oligos were dissolved either in the DEPC water or in human serum. For water samples, the optimized LdCsm detection system produced evident RFU signals in presence of 500 pM-1 nM miR-155 in one hour (Figure 6d), which also constitutes the theoretical detection limit of the system. When miR-155 was dissolved in human serum, a clinic mimic sample, the detection limit was found to 2 nM (Figure 6d). The detection specificity of LdCsm-dCsm3-155-2-A−5U was then evaluated by measuring its detection activity on miR-155 as well as miR-124, miR-149, miR-340, and miR-375, among which the last four human miRNAs are not deregulated in BC tissues [56,57]. Again, miRNA samples were prepared both with the DEPC H_2_O and human sera. miRNA detection with the optimized LdCsm detection system revealed that only miR-155 showed the specific activation of the DNA reporter cleavage, whereas no evident RFU signal was produced in presence of any other miRNAs, including miR-124, miR-149, miR-340, and miR-375 (Figure 6e). 

## 3. Discussion

Type III CRISPR-Cas10 systems possess two target RNA-activated reactions, i.e., the signaling pathway involving the synthesis of the cOA signal in the Palm domain and the HD-hosted collateral DNA cleavage activity [58], both of which can be harnessed for RNA detection. In the signal transduction pathway, the cOA signals synthesized by CRISPR-Cas10 effector complexes specifically activate downstream effectors called CRISPR accessory proteins, many of which exhibit RNase or DNase activity [41,59,60,61,62,63,64]. Thus, they can cleave quenched DNA or RNA fluorescent reporter molecules, yielding fluorescence that can be detected by fluorometric analysis. Several such RNA detection platforms have been established, such as those utilizing *Thermus thermophilus* Csm (TtCsm) and *T. thermophilus* Cmr (TtCmr) systems in conjunction with the *T. thermophilus* Csm6/Csx1 effector proteins [38,39]. In addition, in a new type III-based RNA detection tool, the *Vibrio metoecus* (Vm)Cmr system is combined with VmNucC, a CARF protein derived from the same bacterium that functions in the CBASS antiviral defense [41]. The immune system relies on the target RNA-activated cOA synthesis by VmCmr, producing cA3 that in turn activates VmNucC to cut dsDNA reporters. Furthermore, attempts have also been made to employ both Cas10-hosted activities of the *Lactococcus lactis* (Ll)Csm for RNA detection in a single assay, including the DNase of LlCsm and the RNase of *L. lactis* Csm6 (LlCsm6) that is triggered by cOAs, the secondary signal synthesized by LlCsm. In this case, both ssDNA and RNA reporters are employed [40], while the cOA pathway produces stronger signal and plays a main role in the RNA detection. To date, the LdCsm detection system reported in this work represents the only CRISPR-Cas10 system exploring Cas10 HD domain-mediated DNase for RNA detection. The main feature of the LdCsm-based detection system is that the target NA and reporter NA belong to the different types: while the target of recognition is an RNA, the reporter contains a segment of ssDNA. This provides two advantages: (a) the DNA-based reporters are more stable than the RNA-based ones; and (b) the target RNA is not subjection to destruction by the LdCsm-dCsm3 nuclease, which is indeed the case for the Cas13 nuclease. In addition, while other RNAs in the samples of detection strongly impair the Cas13 detection assay, the influence of non-specific RNAs on the LdCsm RNA detection is greatly reduced. Moreover, we show that the LdCsm effector complex is much more stable than the Cas13 enzyme when stored at the room temperature. This property would increase the self-life and LdCsm detection system’s dependency on cold chain and further increase its application value. These advantages make LdCsm detection a promising candidate for developing simple and reliable RNA detection tools. 

In this article, we explored the potential of LdCsm-dCsm3-based RNA detection platform on detection of miRNA, a putative biomarker for cancer diagnosis. miR-155, a 24 nt breast cancer (BC)-related miRNA, was successfully and specifically detected by the LdCsm detection system, implying that it is a promising candidate for developing miRNA diagnostic tools. To our knowledge, this is the first type III CRISPR-Cas10 system explored for miRNA detection. Theoretical LoD of LdCsm-dCsm3-155-2-A−5U on miR-155 was determined to be 500 pM-1 nM, a value that is comparable to those obtained from the detection of longer RNAs with the TtCsm or TtCmr detection systems without any isothermal amplification [38,39]. The detection limit of miR-155 revealed with the LdCsm platform was about 2 nM in a clinic mimic sample (dissolved in human serum). In comparison, since RNA targets have to be converted to DNA in order to be detected in the qRT-PCR and Cas12 systems, their miRNA detection tools do not work without multiple primer/probe-based amplification reactions [43,45,65,66,67], thus requiring complicated reaction systems. On the other hand, Cas13-based miRNA detection tools are capable of recognizing miRNAs directly in simple reaction systems [46,47,68], but the usage of RNA reporters, which are relatively unstable, may affect their reliability as POC diagnostic tools. Thus, in combination with highly sensitive sensing techniques, like electrochemical measurement techniques, the LdCsm miRNA detection tools could be further developed into the new generations of POC cancer-related miRNA diagnosis tools. 

For type III CRISPR-Cas10-based detection tools, the limited length of miRNAs renders it impossible to choose a completely mismatching 3′-antitag. By contrast, one has to use the 4-5 nt sequence at the 3′-end of miRNAs as the 3′-antitag sequence. We attempted to engineering the repeat sequence of the spacer, generating new 5′-repeat tags that exhibit full mismatches to the 3′-end of miR-155. We show that the complete mismatch between the 5′-repeat tag and the 3′-antitag of miR-155 increased the efficiency of the LdCsm-based miRNA detection system by 3.9-fold (Figure 6). Since a single-base change in the 5′-repeat tag of crRNA does not affect crRNA processing and RNP assembly of LdCsm, we reason that the strategy of repeat engineering may provide a universal solution to similar problems about matching 3′-antitag of target RNAs in other type III-based detection tools.

## 4. Materials and Methods

### 4.1. Bacterial Strains and Growth Conditions

*E. coli* DH5α was the bacterial host for DNA cloning. *E. coli* BL21 (DE3) was the host for expression and purification of LdCsm effector complexes. *E. coli* strains were cultured in Luria–Bertani (LB) or terrific broth (TB) medium. Incubation was at 37 °C, 200 rpm. If required, antibiotics were added as the following: ampicillin (Amp) at 100 μg/mL, kanamycin (Kan) at 25 μg/mL, and chloramphenicol (Cm) at 10 μg/mL.

### 4.2. Construction of Artificial Mini-CRISPR Plasmids Carrying Different Spacers or Repeats

pUCE-X plasmid encodes an artificial mini-CRISPR array carrying an “X” spacer, which transcribes crRNA targeting RNA X specifically. Two spacers named S-155-1 and S-155-2 were designed for targeting breast cancer-related microRNA miR-155. To construct artificial mini-CRISPR fragments of these spacers, fusion PCR amplification was performed using two sets of two primers (miR155-1-F/R, miR155-2-F/R), respectively, to generate multiple copies of Ld repeats interspaced by multiple spacers. mini-CRISPR fragments of about 1 kb were recovered from agarose gel. pUCE plasmid was linearized by *Eco*RI and *Sal*I and then amplified with primers pUCE-repeat-F/R. In this way, the 5′ 18 nt of Ld repeat sequence was added to one end of the pUCE fragment (cut by *Sal*I), while the 3′ 18 nt of the Ld repeat was added to the other end of the pUCE fragment (cut by *EcoR*I). Then, the artificial mini-CRISPR fragments carrying S-155-1/S-155-2 spacer were ligated with pUCE fragments with Ld repeat ends by the Gibson assembly to get pUCE-S-155-1/pUCE-S-155-2 plasmids, which can be used to express crRNA targeting miR-155 specifically. 

Two sets of fusion PCR primers named miR155-1-A−3U-F/R and miR155-2-A−5U-F/R were designed to introduce an A−3U mutation in the Ld repeat sequence of pUCE-S-155-1 and an A−5U mutation in the Ld repeat sequence of pUCE-S-155-2, respectively. The ligation of fusion PCR products of these primers and pUCE fragments with Ld repeat ends produces pUCE-S-155-1-A−3U and pUCE-S-155-2-A−5U, which transcribe crRNA with an A−3U or A−5U mutation in the 5′-repeat tag, respectively. 

### 4.3. Purification of LdCsm Effector Complexes from E. coli

LdCsm-expressing strains were obtained by transformation of three plasmids (p15AIE-Cas-dCsm3, pET30a-Csm2, and pUCE-S1). For purification of each LdCsm effector complex, a few single colonies were inoculated into 20 mL LB medium containing ampicillin, kanamycin, and chloramphenicol and cultivated at 37 °C, 220 rpm overnight. Next, 10 mL of the overnight culture was transferred to 1 L TB medium and cultured to a mid-log phase (OD_600_ = 0.8) under the same growth condition. Then, IPTG was added to 0.3 mM and the culture was further cultured at 25 °C for 16 h to induce the synthesis of LdCsm. Cells were harvested by centrifugation at 5000 rpm for 5 min, and cell pellets were resuspended in 50 mL buffer A (20 mM Tris-HCl, 0.25 M NaCl, 20 mM imidazole, 10% glycerol, pH8.5). The cell suspension was treated by French press at 4 °C, and cell debris was removed by centrifugation at 10,000 rpm for 1 h at 4 °C. The LdCsm complex was then purified in a two-step purification procedure as previously described. 

The LdCsm complex was captured on the HisTrap affinity column and eluted with buffer B (20 mM Tris-HCl, 0.25 M NaCl, 200 mM imidazole, 10% glycerol, and pH 8.5). The product was further purified by gel filtration with Superdex 200 (GE Healthcare, Little Chalfont, United Kingdom) using the chromatography buffer (20 mM Tris-HCl, 0.25 M NaCl, 5% glycerol, and pH 8.5). Fraction samples were collected and mixed thoroughly with glycerol in a volume ratio of 1:1. Purified LdCsm complex samples were then stored at −80 °C or used for further analysis.

### 4.4. Collateral Nucleic Acid Cleavage Assay

RNA-activated ssDNA cleavage was assayed in 10 μL reactions in the cleavage buffer (50 mM Tris-HCl, 10 mM MgCl_2_, 50 mM KCl, 0.1 mg/mL BSA, and pH 7.0) containing 50 nM LdCsm complex, 50 nM 5′ FAM-labeled ssDNA substrate, and 500 nM target RNA or unspecific RNA. Nucleic acid cleavage was conducted at 37 °C for 10 min and stopped by addition of 2 × RNA loading dye (New England Biolabs, Ipswich, MA, USA). Before loading, samples were heated for 3 min at 95 °C and analyzed by denaturing polyacrylamide gel electrophoresis. Cleavage products were visualized by fluorescence imaging with xx.

### 4.5. Evaluation of Fluorescence DNA Reporter Cleavage Assay/RNA Detection Reaction

DNA reporter reactions were prepared in a total of 20 μL in the cleavage buffer as described above, in which 1 μM Poly-T or CT ssDNA reporter was used as the ssDNA substrate (Sequences listed in Appendix A), whereas target RNA was the RNA sample prepared with DEPC H_2_O or serum. Addition of one of the ssDNA reporters and one of the RNA samples to be detected yielded individual detection reactions, which were set up in a 384-well black plate (Thermo fisher, Waltham, MA, USA). The plate was then placed in a fluorescence plate reader (Perkin Elmer, Waltham, MA, USA) and incubated for 30–60 min at 37 °C, during which the fluorescence signal (relative fluorescence unit, RFU) was measured for each well every 5/10 min (λ_ex_: 485 nm; λ_em_: 535 nm). Background RFU values generated for the reference sample (lacking a target RNA) were subtracted from the RFU values obtained from the reactions containing the cognate target RNA or RNA samples to be detected. The yielding background-subtracted RFU values were used for calculating the RFU generation rate in each reaction, which represented the DNase activity of activated-LdCsm RNP complex or the detection signal of LdCsm system on the indicated RNA samples.

### 4.6. RNA Detection Reaction Using Cas13a

The Cas13a detection kit contained the LwaCas13a protein (catalogue number: M20202-0100, 100 pmol/100 μL), a fluorescence-quenched RNA reporter (5′ FAM-labeled and 3′ BHQ1-labeled, sequence listed in Appendix A), and 10× reaction buffer (purchased from the Bio-lifesci company (Guangzhou, China)). The storage buffer of the LwaCas13a protein contains 20 mM Tris-HCl (pH 7.4), 0.1 mM EDTA, 1 mM DTT, 200 mM NaCl, and 50% (*v*/*v*) glycerol. A Cas13a crRNA targeting S1 target RNA (sequence listed in Appendix A) was ordered from the same company. The detection reaction (20 μL) contains 1× reaction buffer (10 mM Tris-HCl (pH 8.3), 50 mM KCl, 1.5 mM MgCl_2_), 100 nM LwaCas13a, 100 nM Cas13a crRNA, 1 μM RNA reporter, and an indicated concentration of target RNA or unspecific RNA. After the addition of the RNAs to be detected (target RNA), the detection reactions were set up in a 384-well black plate (Thermo fisher, Waltham, MA, USA) and placed in a fluorescence plate reader (Perkin Elmer, Waltham, MA, USA) to measure fluorescence signals. The detection activity was calculated as by the LdCsm detection reaction described above. 

## 5. Conclusions

An RNA detection platform based on the LdCsm-dCsm3 system was developed, which solely relies on its target RNA-activated DNase to generate fluorescence from a quenched DNA reporter. Compared to Cas13a detection system, this LdCsm-dCsm3-based detection system has the advantage of tolerance against disturbance from unspecific RNA and shows less dependency on the cold chain transportation. The detection system showed high detection specificity on target miRNA since no fluorescent signal was generated in presence of other miRNAs. The LoD determined for the breast cancer-related miR-155 was 2 nM in clinic mimic samples, and introducing a single-base mutation into the 5′-repeat tag of LdCsm crRNA increased the detection activity by 3.9-fold. Therefore, the LdCsm-dCsm3 system represents a promising candidate for developing a new generation of point-of-care miRNA diagnostic tools. 

## 6. Patents

A patent application was filed for optimization of LdCsm DNase-based RNA detection system and for its application in harnessing the optimized system for RNA detection.

## Figures and Tables

**Figure 1 ijms-24-02857-f001:**
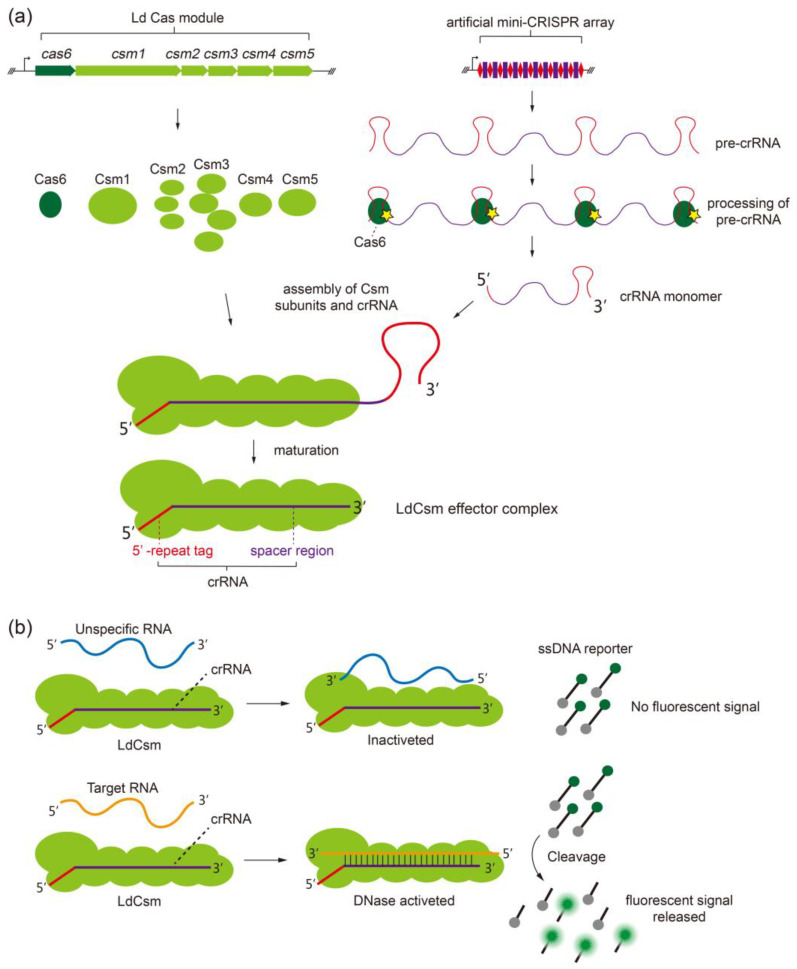
Rationale of the LdCsm-based RNA detection. (**a**) Schematic of the expression of the LdCsm system and the assembly of the LdCsm effector RNP complex. (**b**) Schematic of the LdCsm-based RNA detection reaction. Target RNA binds to the LdCsm RNP complex and activates the collateral DNase hosted by the Cas10 HD domain, an activity that cleaves the fluorescence-quenched ssDNA reporter (5′-FAM labeled and 3′-BHQ1 labeled ssDNA substrate), releasing fluorescence signals to be detected and recorded by fluorometry. RFU: relative fluorescence units.

**Figure 2 ijms-24-02857-f002:**
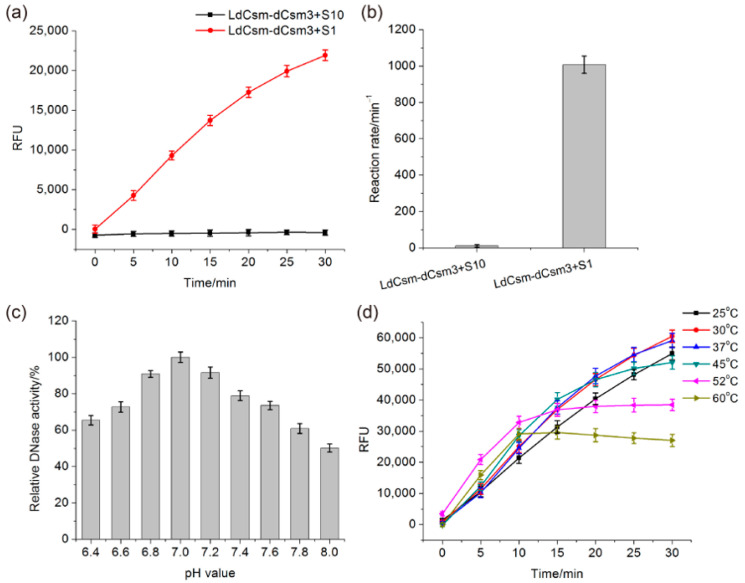
Testing the LdCsm-dCsm3 RNA detection system. (**a**) Time course experiment. (**b**) Relative enzyme activity in (**a**). (**c**) Effect of pH values on the LdCsm-dCsm3 DNase. The activity at pH 7.0 was set as 100%. (**d**) Effect of temperatures on the LdCsm-dCsm3 DNase. RNAs to be detected were added to 50 nM in the reactions. RFU: relative fluorescence units.

**Figure 3 ijms-24-02857-f003:**
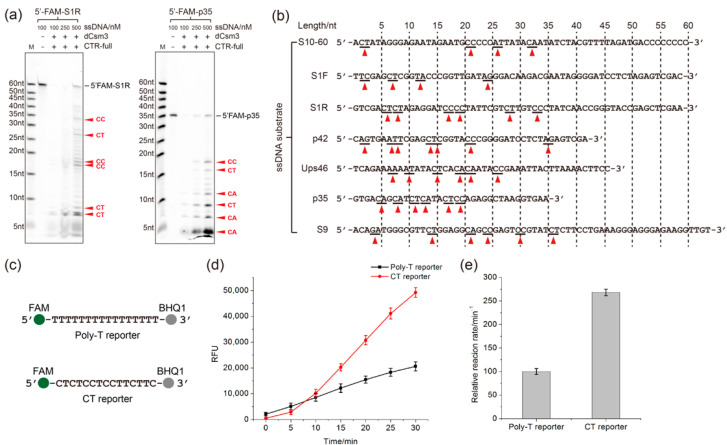
Determination of the nucleotide preference of the LdCsm DNase and testing of the optimized ssDNA reporter. (**a**) PAGE analyses of the preference of DNA cleavage by the LdCsm-dCsm3 DNase. Two ssDNA substrates were incubated with LdCsm-dCsm3, and cleavage products were analyzed by the denaturing PAGE. Bands of main cleavage products are indicated by red triangles. Names of the main dinucleotide cleavage sites are shown on the right of the cleaved products. (**b**) Summary of LdCsm cleavage sites on seven ssDNA substrates revealed from the denaturing PAGE, which are indicated by red triangles underneath their sequences. The sites predicted to be preferred by LdCsm-dCsm3 DNase are labeled by red triangles. (**c**) Schematic of Poly-T reporter and the CT reporter. (**d**) RNA detection reaction by the LdCsm-dCsm3 system using each DNA reporter. (**e**) Relative detection efficiency converted from the data in (**d**) in which the activity with Poly-T reporter was set as 100%.

**Figure 4 ijms-24-02857-f004:**
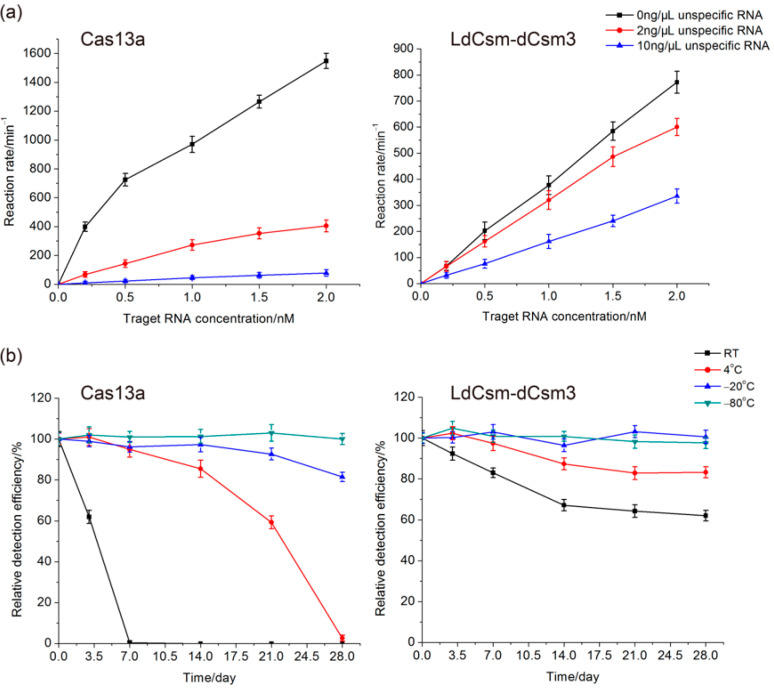
Comparison of the LwaCas13 and LdCsm-dCsm3 detection systems. (**a**) Influence of unspecific RNAs. (**b**) Influence of storage conditions. Detection efficiency of Cas13a or LdCsm-dCsm3 stored for 0 days was set as 100% each. The detection assay was set up with 100 nM Cas enzymes (Cas13a or LdCsm) and 10 μM RNA/DNA reporter. Unspecific RNAs were total RNAs prepared from *E. coli* BL21 (DE3) cells.

**Figure 5 ijms-24-02857-f005:**
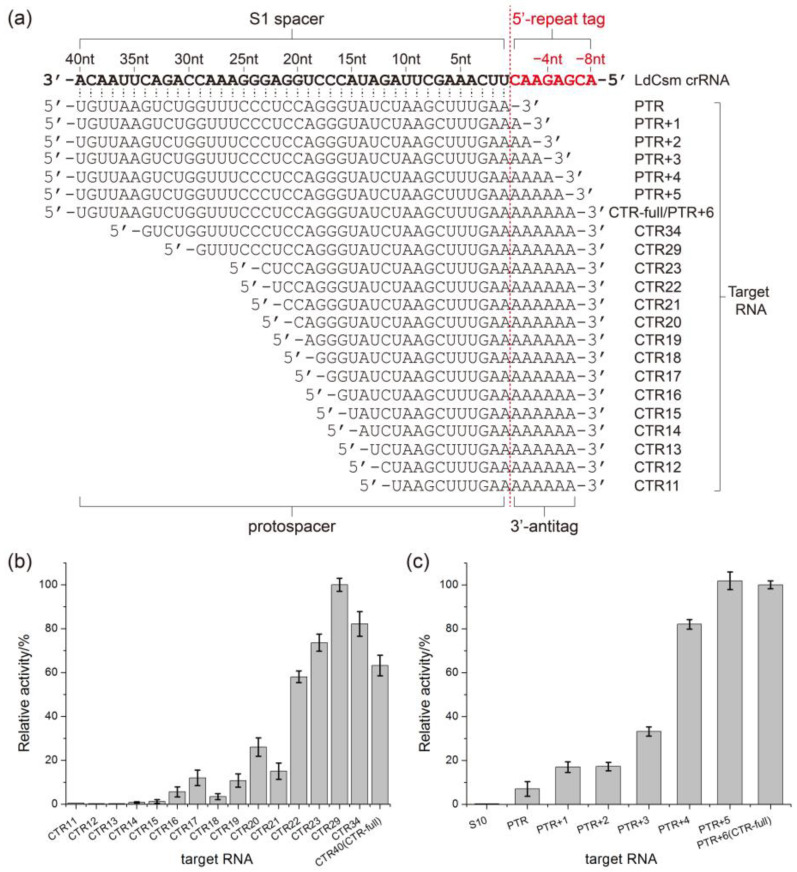
Minimal length of the activator RNA required for the LdCsm-dCsm3 collateral DNase. (**a**) Schematic of designed activator RNAs. The spacer sequence of the crRNA is in bold whereas the 8-nt 5′-repeat tag is highlighted in red. (**b**) Detection activity of LdCsm-dCsm3 in the presence of target RNAs of different lengths. The length of each CTR is indicated with the number of nucleotide sequence that are complementary to the spacer of crRNA. Relative activity of the reaction treated by CTR29 is set as 100%. (**c**) Detection activity of LdCsm-dCsm3 activated by target RNAs carrying various lengths of the 3′-antitag. S10 is an unspecific RNA, serving as a negative control. Protospacer target RNA (PTR) is the target RNA without any 3′-antitag sequence. Relative activity of the reaction treated by CTR-full is set as 100%.

**Figure 6 ijms-24-02857-f006:**
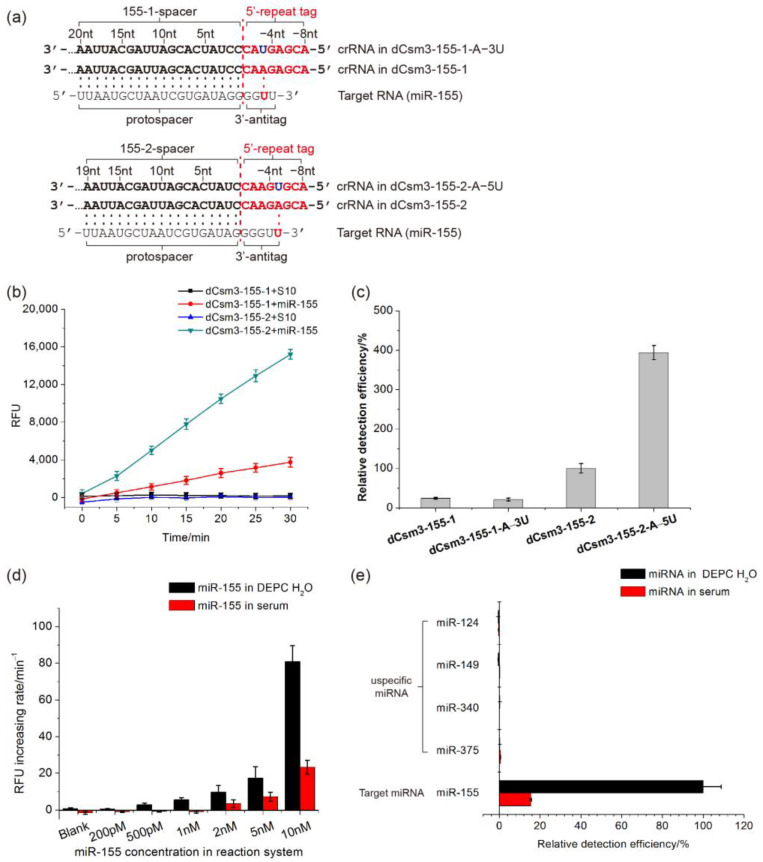
Detection of miR-155 by using the LdCsm-dCsm3 detection system. (**a**) Schematic of the two crRNAs designed for miR-155 detection (155-1 and 155-2) and their repeat-mutated derivatives (155-1-A−3U and 155-2-A−5U). 5′-repeat tags of crRNAs are highlighted in red. Base pairings between 3′-antitag of miR-155 and the 5′-repeat tag of each crRNA are highlighted in red; mutated bases in the repeat tags are shown in dark blue. (**b**) Time course experiment of miR-155 detection by LdCsm-dCsm3 carrying crRNAs derived from the 155-1 or 155-2 spacer (denoted as dCsm3-155-1 or dCsm3-155-2, respectively). (**c**) Relative detection activity of four LdCsm-dCsm3 effectors carrying each of the four designed spacers. Detection activity of dCsm3-155-2 was set as 100%. (**d**) Specific detection of miR-155 in clinic mimic samples. (**e**) Specificity of dCsm3-155-2-A−5U. Target miRNA miR-155 or unspecific miRNAs including miR-124, miR-149, miR-340, and miR-375 (sequences listed in Appendix A) were each dissolved in the DEPC H_2_O or in human sera. The final concentration of miRNAs in these reaction systems is 10 nM.

## Data Availability

All other data generated during the study are available from the corresponding author upon reasonable request.

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
