# Peer review of "Harnessing the LdCsm RNA Detection Platform for Efficient microRNA Detection"

_ijms, 2023, doi:10.3390/ijms24032857_

Round 1

Reviewer 1 Report

In this manuscript, Yu et al. established a miRNA detection platform based on the powerful targeting of RNA-activated DNase activity of type III-A Csm system. Through probing and condition optimization, the platform enabled the detection of miRNA-155 associated with human breast cancer, which is expected to be a point-of-care diagnostic tool. The data included mostly supported the conclusion drawn by the authors. The work may be publishable on International Journal Of Molecular Sciences after the following concerns are addressed.

1.     In Figure 2a, the authors used target RNA S1 and non-homologous RNA S10 to verify the possibility of specific detection of RNA based on the LdCsm-dCsm3 system. The reaction time in the design was only 30 min, but the plateau period was not reached at 30 min. Please explain.

2.     In the experimental design, do the ions often added to buffer affect LdCsm-dCsm3 DNase enzyme activity? Please provide relevant data and explain.

3.     In Figure 3a, b, the authors claim that “Major cleavage sites were then deduced according to the sizes, and this revealed that the preferred motifs include CT, CC and CA sites”, but more sufficient evidences are expected to support the conclusion.

4.      In Figure 4a, there were significant differences in the vertical coordinates between the Cas13a-based and LdCsm-based detection system, especially in the "0 ng/uL unspecific RNA" group. Was this phenomenon due to the stronger effect of Cas13a on target recognition than dCsm3? Please explain the phenomenon, maybe from the perspective of the binding mechanism.

Author Response

Response to reviewer 1

Comments:

In this manuscript, Yu et al. established a miRNA detection platform based on the powerful targeting of RNA-activated DNase activity of type III-A Csm system. Through probing and condition optimization, the platform enabled the detection of miRNA-155 associated with human breast cancer, which is expected to be a point-of-care diagnostic tool. The data included mostly supported the conclusion drawn by the authors. The work may be publishable on International Journal Of Molecular Sciences after the following concerns are addressed.

Reply: We highly appreciate the positive comments from this reviewer as well as the insightful suggestions listed below, which greatly help us in revision of this manuscript.

Point-to-point responses:

1. In Figure 2a, the authors used target RNA S1 and non-homologous RNA S10 to verify the possibility of specific detection of RNA based on the LdCsm-dCsm3 system. The reaction time in the design was only 30 min, but the plateau period was not reached at 30 min. Please explain.

Reply: The time required for reaching the plateau period of LdCsm RNA detection reactions is influenced by the concentrations of three reagents in the reaction: the ssDNA reporter molecules, the LdCsm-dCsm3 effector and the target RNA. Since the first two factors are constant in all reactions, the slope of the RFU increase during the linear increase period well reflects the concentration of the RNA to be detected whereas using the final RFU values at the plateau period will underestimate the RNA concentration if either the reporter or the LdCsm effector or both become the limiting factor(s) of the RNA detection reaction. For determination of the increase rate of RFU, we found a reaction time of 30 min is sufficient for the accurate detection of target RNA in our assay.

2. In the experimental design, do the ions often added to buffer affect LdCsm-dCsm3 DNase enzyme activity? Please provide relevant data and explain.

Reply: Yes, the LdCsm DNase does require Mg++ metal ion and the relevant data have been reported in the Supplementary Figure 4d in ref [48] (Lin et al., 2020, NAR, our previous work). We have indicated this by amending the following in the Results part: “The metal ion dependence of the LdCsm DNase has been tested in our previous work where we found that the LdCsm DNase is activated by 10 mM MgCl2 and addition of 50 mM KCl further increases the DNA cleavage activity (). Here, ...” in Line 139-141. Thanks for the suggestion.

3. In Figure 3a, b, the authors claim that “Major cleavage sites were then deduced according to the sizes, and this revealed that the preferred motifs include CT, CC and CA sites”, but more sufficient evidences are expected to support the conclusion.

Reply: Thanks for pointing out the weakness of our data analysis. We have now re-analyzed that data of ssDNA cleavage, revised the figures showing the cleavage sites (Figure 3a in the revised manuscript, Figure S1 in the revised supplementary files) and summarized the cleavage sites into a table (Table S2 in the revised supplementary files). We then revised data presentation in the Results part accordingly. This part is now read as the following: “......seven different 5’ FAM-labelled ssDNAs (sequences listed in Table S1) were determined for main cleavage sites of LdCsm-dCsm3 by polyacrylamide gel electrophoresis (PAGE) analyses of trans-cleavage products (Figure 3a, Figure S1). Main cleavage sites were then deduced for each ssDNA substrate based on the sizes of cleaved products of a high abundance (Figure 3b). This revealed a total of 39 main cleaved sites of 10 different dinucleotides including 11 CT, 9 CC and 7 CA dinucleotides, whose sum accounts for 69% of the total number (Table S2). Moreover, CTs are among the main cleavage sites in all 7 ssDNA substrates, while CC and CA occurred in 6 and 4 of them, respectively (Table S2). Therefore, we concluded that CT, CC and CA dinucleotides are preferred ssDNA sites of the LdCsm-dCsm3 DNase.”

4.  In Figure 4a, there were significant differences in the vertical coordinates between the Cas13a-based and LdCsm-based detection system, especially in the "0 ng/uL unspecific RNA" group. Was this phenomenon due to the stronger effect of Cas13a on target recognition than dCsm3? Please explain the phenomenon, maybe from the perspective of the binding mechanism.

Reply: According to our result, without disturbance from unspecific RNA, Cas13a detection system showed higher RFU increasing rates than LdCsm-dCsm3 detection system on equal concentration of target RNA. Clearly, upon activation, the Cas13a RNase cleaves RNA reporter at a higher rate than the LdCsm-dCsm3 DNase does the ssDNA reporter. Possibly, the higher relative activity of Cas13a RNase could be attributed to higher efficiency in capturing NA substrates and in releasing cleaved products from their active sites relative to the LdCsm DNase. However, we prefer not to make any speculation on these possible mechanisms since this work is not aimed to investigate the involved mechanism of NA cleavage for any of the RNA-guided nucleases. Thus, such a discussion could be considered as out of the scope of the current article and we will turn back to these important issues in our future research.

Reviewer 2 Report

This manuscript from Yu and colleagues presents a thorough and rigorous investigation showing that the the LdCsm platform has great potential for  miRNA detection. The experimental plan is logical and the results clearly presented, and the conclusions are well supported.

However, there are several limitations in the study and some improvements that should be made, as follows:

1. How dose the LdCsm platform perform compared with the traditional qRT-PCR on miRNA detection, is the sensitivity better, is the workflow more straightforward? 

2.In Figure 2(a), the max incubation time is 30 minutes, what about beyond 30 minutes, is the fluorescence signal still increasing?

3.Figure 4(a), besides E. Coli RNA, did the author test human RNA sample? 

4. A lot of the statements in this manuscript, especially the introduction part, missing the relevant references. Please add the ref.

5.line 64. author should state what cOA stands for if appeared 1st time in the manuscript.

Author Response

Response to reviewer 2

Comments:

This manuscript from Yu and colleagues presents a thorough and rigorous investigation showing that the the LdCsm platform has great potential for miRNA detection. The experimental plan is logical and the results clearly presented, and the conclusions are well supported.

Reply: We highly appreciate the positive comments from this reviewer as well as the insightful suggestions listed below, which greatly help us in revision of this manuscript.

Point-to-point responses:

However, there are several limitations in the study and some improvements that should be made, as follows:

1. How dose the LdCsm platform perform compared with the traditional qRT-PCR on miRNA detection, is the sensitivity better, is the workflow more straightforward? 

Reply: The LdCsm detection procedure is more straightforward compared with qRT-PCR and Cas12. We have now pointed this out in the Discussion part. Thanks for the suggestion.

2. In Figure 2(a), the max incubation time is 30 minutes, what about beyond 30 minutes, is the fluorescence signal still increasing?

Reply: The time required for reaching the plateau period of LdCsm RNA detection reactions is influenced by the concentrations of three reagents in the reaction: the ssDNA reporter molecules, the LdCsm-dCsm3 effector and the target RNA. Since the first two factors are constant in all reactions, the slope of the RFU increase during the linear increase period well reflects the concentration of the RNA to be detected whereas using the final RFU values at the plateau period will underestimate the RNA concentration if either the reporter or the LdCsm effector or both become the limiting factor(s) of the RNA detection reaction. For determination of the increase rate of RFU, we found a reaction time of 30 min is sufficient for the accurate detection of target RNA in our assay.

3. Figure 4(a), besides E. Coli RNA, did the author test human RNA sample? 

Reply: Human RNA sample has not been tested. Although one could argue on that using a human RNA sample is more relevant, however, in the experimental setup here, pure RNAs of any source would serve the same function, i.e., to test the tolerance of the two detection systems to unspecific RNAs. In our next experiments, both systems will be employed to test with clinical samples to reveal their usefulness in miRNA detection.

4. A lot of the statements in this manuscript, especially the introduction part, missing the relevant references. Please add the ref.

Reply: Relevant references have been added in introduction and discussion of the revised manuscript, and thanks for pointing out the problem.

5. line 64. author should state what cOA stands for if appeared 1st time in the manuscript.

Reply: Illustration of cOA (cyclic oligoadenylate) has been added to line 64 in the revised manuscript.

Round 2

Reviewer 1 Report

The authors have addressed my concerns. I think it can be accepted.